# LEARNING GAUSSIAN POLICIES FROM SMOOTHED ACTION VALUE FUNCTIONS

## ABSTRACT

State-action value functions (*i.e.,* Q-values) are ubiquitous in reinforcement learning (RL), giving rise to popular algorithms such as SARSA and Q-learning. We propose a new notion of action value defined by a Gaussian smoothed version of the expected Q-value. We show that such smoothed Q-values still satisfy a Bellman equation, making them learnable from experience sampled from an environment. Moreover, the gradients of expected reward with respect to the mean and covariance of a parameterized Gaussian policy can be recovered from the gradient and Hessian of the smoothed Q-value function. Based on these relationships we develop new algorithms for training a Gaussian policy directly from a learned smoothed Q-value approximator. Our approach is amenable to proximal optimization techniques by augmenting the objective with a penalty on KL-divergence from a previous policy. We find that the ability to learn both a mean and covariance during training allows this approach to achieve much better results on standard continuous control benchmarks.

## 1 INTRODUCTION

Model-free reinforcement learning algorithms often alternate between two concurrent but interacting processes: (1) *policy evaluation*, where an *action value function* (*i.e.,* a Q-value) is updated to obtain a better estimate of the return associated with taking a specific action, and (2) *policy improvement*, where the policy is updated aiming to maximize the current value function. In the past, different notions of Q-value have led to distinct but important families of RL methods. For example, SARSA (Rummery & Niranjan, 1994; Sutton & Barto, 1998; Van Seijen et al., 2009) uses the *expected* Q-value, defined as the expected return of following the current policy. Q-learning (Watkins, 1989) exploits a *hard-max* notion of Q-value, defined as the expected return of following an optimal policy. Soft Q-learning (Haarnoja et al., 2017) and PCL (Nachum et al., 2017a) both use a *soft-max* form of Q-value, defined as the future return of following an optimal entropy regularized policy. Clearly, the choice of Q-value function has a considerable effect on the resulting algorithm; for example, restricting the types of policies that can be expressed, and determining the type of exploration that can be naturally applied.

In this work we introduce a new notion of action value: the *smoothed* action value function $\tilde{Q}^\pi$. Unlike previous notions, which associate a value with a specific action at each state, the smoothed Q-value associates a value with a specific *distribution* over actions. In particular, the smoothed Q-value of a state-action pair $(s, a)$ is defined as the expected return of first taking an action sampled from a normal distribution $N(a, \Sigma(s))$, centered at $a$, then following actions sampled from the current policy thereafter. In this way, the smoothed Q-value can also be interpreted as a Gaussian-smoothed or noisy version of the expected Q-value.

We show that smoothed Q-values possess a number of interesting properties that make them attractive for use in RL algorithms. For one, the smoothed Q-values satisfy a single-step Bellman consistency, which allows bootstrapping to be used to train a function approximator. Secondly, for Gaussian policies, the standard optimization objective (expected return) can be expressed in terms of smoothed Q-values. Moreover, the gradient of this objective with respect to the mean and covariance of the Gaussian policy is equivalent to the gradient and the Hessian of the smoothed Q-value function, which allows one to derive updates to the policy parameters by having access to the derivatives of a sufficiently accurate smoothed Q-value function.

This observation leads us to propose an algorithm called *Smoothie*, which in the spirit of (Deep) Deterministic Policy Gradient (DDPG) (Silver et al., 2014; Lillicrap et al., 2016), trains a policy using the derivatives of a trained (smoothed) Q-value function, thus avoiding the high-variance of stochastic updates used in standard policy gradient algorithms (Williams & Peng, 1991; Konda & Tsitsiklis, 2000). Unlike DDPG, which is well-known to have poor exploratory behavior (Haarnoja et al., 2017), the approach we develop is able to utilize a non-deterministic Gaussian policy parameterized by both a mean and a covariance, thus allowing the policy to be exploratory by default and alleviating the need for excessive hyperparameter tuning.

Furthermore, we show that Smoothie can be easily adapted to incorporate proximal policy optimization techniques by augmenting the objective with a penalty on KL-divergence from a previous version of the policy. The inclusion of a KL-penalty is not feasible in the standard DDPG algorithm, but we show that it is possible with our formulation, and it significantly improves stability and overall performance. On standard continuous control benchmarks, our results are competitive with or exceed state-of-the-art, especially for more difficult tasks in the low-data regime.

## 2 NOTATION & BACKGROUND

We consider the standard model-free RL framework, where an agent interacts with a stochastic black-box environment by sequentially observing the state of the environment, emitting an action, and receiving a reward feedback; the goal is to find an agent that achieves maximal cumulative discounted reward. This problem can be expressed in terms of a Markov decision process (MDP) that consists of a state space $\mathcal{S}$ and an action space $\mathcal{A}$, where at iteration $t$ the agent encounters a state $s_t \in \mathcal{S}$ and emits an action $a_t \in \mathcal{A}$, after which the environment returns a scalar reward $r_t \sim R(s_t, a_t)$ and places the agent in a new state $s_{t+1} \sim P(s_t, a_t)$.

We model the behavior of the agent using a stochastic policy $\pi$ that produces a distribution over feasible actions at each state $s$ as $\pi(a \mid s)$. The optimization objective (expected discounted return), as a function of the policy, can then be expressed in terms of the expected action value function $Q^\pi(s, a)$ by,

$$O_{\text{ER}}(\pi) \;=\; \int_{\mathcal{S}} \int_{\mathcal{A}} \pi(a \mid s) Q^\pi(s, a) \, \mathrm{d}a \, \mathrm{d}\rho^\pi(s) \;, \tag{1}$$

where $\rho^\pi(s)$ is the stationary distribution of the states under $\pi$, and $Q^\pi(s, a)$ is recursively defined using the Bellman equation,

$$Q^\pi(s, a) = \mathbb{E}_{r, s'} \left[ r + \gamma \int_{\mathcal{A}} Q^\pi(s', a') \pi(a' \mid s') \, \mathrm{d}a' \right] \;, \tag{2}$$

where $\gamma \in [0, 1]$ is the discount factor. For brevity, we will often suppress explicit denotation of the sampling distribution $R$ over immediate rewards and the distribution $P$ over state transitions.

The policy gradient theorem (Sutton et al., 2000) expresses the gradient of $O_{\text{ER}}(\pi_\theta)$ *w.r.t.* $\theta$, the tunable parameters of a policy $\pi_\theta$, as,

$$\nabla_\theta O_{\text{ER}}(\pi_\theta) \;=\; \int_{\mathcal{S}} \int_{\mathcal{A}} \nabla_\theta \pi_\theta(a \mid s) Q^\pi(s, a) \, \mathrm{d}a \, \mathrm{d}\rho^\pi(s) \tag{3}$$

$$=\; \int_{\mathcal{S}} \mathbb{E}_{a \sim \pi_\theta(a \mid s)} \left[ \nabla_\theta \log \pi_\theta(a \mid s) Q^\pi(s, a) \right] \mathrm{d}\rho^\pi(s) \;. \tag{4}$$

Many reinforcement learning algorithms, including policy gradient and actor-critic variants, trade off variance and bias when estimating the random variable inside the expectation in (4); for example, by attempting to estimate $Q^\pi(s, a)$ accurately using function approximation. In the simplest scenario, an unbiased estimate of $Q^\pi(s, a)$ is formed by accumulating discounted rewards from each state forward using a single Monte Carlo sample.

In this paper, we focus on multivariate Gaussian policies over continuous action spaces, $\mathcal{A} \equiv \mathbb{R}^{d_a}$. We represent the observed state of the MDP as a $d_s$-dimensional feature vector $\Phi(s) \in \mathbb{R}^{d_s}$, and parametrize the Gaussian policy by a mean and covariance function, respectively $\mu(s) : \mathbb{R}^{d_s} \to \mathbb{R}^{d_a}$ and $\Sigma(s) : \mathbb{R}^{d_s} \to \mathbb{R}^{d_a} \times \mathbb{R}^{d_a}$. These map the observed state of the environment to a Gaussian distribution,

$$\pi(a \mid s) \;=\; N(a \mid \mu(s), \Sigma(s)) \;=\; |2\pi\Sigma(s)|^{-1/2} \exp\left\{ -\frac{1}{2} \|a - \mu(s)\|^2_{\Sigma(s)^{-1}} \right\}, \tag{5}$$

where $\|v\|_A^2 = v^{\mathsf{T}} A v$. Below we develop new RL training methods for this family of parametric policies, but some of the ideas presented may generalize to other families of policies as well. We begin the formulation by reviewing some prior work on learning Gaussian policies.

## 2.1 Deterministic Policy Gradient

Silver et al. (2014) present a new formulation of the policy gradient, called the *deterministic policy gradient*, for the family of Gaussian policies in the limit where the policy covariance approaches zero. In such a scenario, the policy becomes deterministic because sampling from the policy always returns the Gaussian mean. The key observation of (Silver et al., 2014) is that under a deterministic policy $\pi \equiv (\mu, \Sigma \to 0)$, one can estimate the expected future return from a state $s$ as,

$$\lim_{\Sigma \to 0} \int_{\mathcal{A}} \pi(a \mid s) Q^\pi(s, a) \, \mathrm{d}a \;=\; Q^\pi(s, \mu(s)) \;. \tag{6}$$

Then, one can express the gradient of the optimization objective (expected discounted return) for a parameterized $\pi_\theta \equiv \mu_\theta$ as,

$$\nabla_\theta O_{\text{ER}}(\pi_\theta) \;=\; \int_{\mathcal{S}} \nabla_\theta Q^\pi(s, \mu_\theta(s)) \mathrm{d}\rho^\pi(s) \;=\; \int_{\mathcal{S}} \frac{\partial Q^\pi(s, a)}{\partial a}\Big|_{a = \mu_\theta(s)} \nabla_\theta \mu_\theta(s) \mathrm{d}\rho^\pi(s) \;. \tag{7}$$

This can be thought of as a characterization of the policy gradient theorem for deterministic policies.

In the limit of $\Sigma \to 0$, one can also re-express the Bellman equation (2) as,

$$Q^\pi(s, a) \;=\; \mathbb{E}_{r, s'} \left[ r + Q^\pi(s', \mu(s')) \right] \;. \tag{8}$$

Therefore, a value function approximator $Q_w^\pi$ can be optimized by minimizing the Bellman error,

$$E(w) = \sum_{(s, a, r, s') \in \mathcal{D}} (Q_w^\pi(s, a) - r - \gamma Q_w^\pi(s', \mu(s')))^2 \;, \tag{9}$$

for transitions $(s, a, r, s')$ sampled from a dataset $\mathcal{D}$ of interactions of the agent with the environment. Algorithms like DDPG (Lillicrap et al., 2016) alternate between improving the value function by gradient descent on (9), and improving the policy based on (7).

In practice, to gain better sample efficiency, Degris et al. (2012) and Silver et al. (2014) replace the on-policy state distribution $\rho^\pi(s)$ in (7) with an off-policy distribution $\rho^\beta(s)$ based on a replay buffer. After this substitution, the policy gradient identity in (7) does not hold exactly, however, prior work finds that this works well in practice and improves sample efficiency. We also adopt a similar approximation in our method to make use of off-policy data.

## 3 Smoothed Action Value Functions

In this paper, we introduce *smoothed action value functions*, the gradients of which provide an effective signal for optimizing the parameters of a Gaussian policy. Our notion of smoothed Q-values, denoted $\tilde{Q}^\pi(s, a)$, differs from ordinary Q-values $Q^\pi(s, a)$ in that smoothed Q-values do not assume the first action of the agent is fully specified, but rather they assume that only the mean of the distribution of the first action is known. Hence, to compute $\tilde{Q}^\pi(s, a)$, one has to perform an expectation of $Q^\pi(s, \tilde{a})$ for actions $\tilde{a}$ drawn in the vicinity of $a$. More formally, smoothed action values are defined as,

$$\tilde{Q}^\pi(s, a) \;=\; \int_{\mathcal{A}} N(\tilde{a} \mid a, \Sigma(s)) \, Q^\pi(s, \tilde{a}) \mathrm{d}\tilde{a} \;. \tag{10}$$

With this definition of $\tilde{Q}^\pi$, one can re-express the expected reward objective for a Gaussian policy $\pi \equiv (\mu, \Sigma)$ as,

$$O_{\text{ER}}(\pi) = \int_{\mathcal{S}} \tilde{Q}^\pi(s, \mu(s)) \mathrm{d}\rho^\pi(s). \tag{11}$$

The insight that differentiates this approach from prior work including Heess et al. (2015); Ciosek & Whiteson (2017) is that instead of learning a function approximator for $Q^\pi(s, a)$ and then drawing samples to approximate the expectation in (10) and its derivative, we directly learn a function approximator for $\tilde{Q}^\pi(s, a)$.

The key observation that enables direct bootstrapping of smoothed Q-values, $\tilde{Q}^\pi(s, a)$, is that their form allows a notion of Bellman consistency. First, note that for Gaussian policies $\pi \equiv (\mu, \Sigma)$ we have

$$Q^\pi(s, a) = \mathbb{E}_{r,s'}[r + \gamma \tilde{Q}^\pi(s', \mu(s'))] . \tag{12}$$

Then, combining (10) and (12), one can derive the following one-step Bellman equation for smoothed Q-values,

$$\tilde{Q}^\pi(s, a) = \int_{\mathcal{A}} N(\tilde{a} \mid a, \Sigma(s)) \, \mathbb{E}_{\tilde{r},\tilde{s}'} \left[ \tilde{r} + \gamma \tilde{Q}^\pi(\tilde{s}', \mu(\tilde{s}')) \right] \mathrm{d}\tilde{a} , \tag{13}$$

where $\tilde{r}$ and $\tilde{s}'$ are sampled from $R(s, \tilde{a})$ and $P(s, \tilde{a})$. Below, we elaborate on how one can make use of the derivatives of $\tilde{Q}^\pi$ to learn $\mu$ and $\Sigma$, and how the Bellman equation in (13) enables direct optimization of $\tilde{Q}^\pi$.

## 3.1 Policy Improvement - Optimizing $(\mu_\theta, \Sigma_\phi)$ Given $\tilde{Q}^\pi$

We parameterize a Gaussian policy $\pi_{\theta,\phi} \equiv (\mu_\theta, \Sigma_\phi)$ in terms of two sets of parameters $\theta$ and $\phi$ for the mean and the covariance. The gradient of the objective *w.r.t.* mean parameters follows from the policy gradient theorem and is almost identical to (7),

$$\nabla_\theta O_{\text{ER}}(\pi_{\theta,\phi}) = \int_{\mathcal{S}} \frac{\partial \tilde{Q}^\pi(s, a)}{\partial a} \Big|_{a=\mu_\theta(s)} \nabla_\theta \mu_\theta(s) \mathrm{d}\rho^\pi(s). \tag{14}$$

Estimating the derivative of the objective *w.r.t.* covariance parameters is not as straightforward, since $\tilde{Q}^\pi$ is not a direct function of $\Sigma$. However, a key observation of this work is that the second derivative of $\tilde{Q}^\pi$ *w.r.t.* actions is sufficient to exactly compute the derivative of $\tilde{Q}^\pi$ *w.r.t.* $\Sigma$,

$$\frac{\partial \tilde{Q}^\pi(s, a)}{\partial \Sigma(s)} = \frac{1}{2} \cdot \frac{\partial^2 \tilde{Q}^\pi(s, a)}{\partial a^2}. \tag{15}$$

A proof of this identity is provided in the Appendix. The proof may be easily derived by expressing both sides of the equation using standard matrix calculus like $\frac{\partial}{\partial A}|A|^{-1/2} = -\frac{1}{2}|A|^{-1/2}A^{-1}$ and $\frac{\partial}{\partial A}||v||^2_{A^{-1}} = -A^{-1}vv^T A^{-1}$.

Then, the full derivative *w.r.t.* $\phi$ takes the form,

$$\nabla_\phi O_{\text{ER}}(\pi_{\theta,\phi}) = \frac{1}{2} \int_{\mathcal{S}} \frac{\partial^2 \tilde{Q}^\pi(s, a)}{\partial a^2} \Big|_{a=\mu_\theta(s)} \nabla_\phi \Sigma_\phi(s) \mathrm{d}\rho^\pi(s). \tag{16}$$

## 3.2 Policy Evaluation - Optimizing $\tilde{Q}^\pi_w$ Given $(\mu, \Sigma)$

We can think of two ways to optimize $\tilde{Q}^\pi_w$. The first approach leverages (10) to update $\tilde{Q}^\pi$ based on expected Q-value function $Q^\pi$. In such an approach, one trains a parameterized $\tilde{Q}^\pi_w$ to approximate the standard expected Q-value function $Q^\pi$ using standard methods (see *e.g.,* Rummery & Niranjan (1994); Sutton & Barto (1998); Van Seijen et al. (2009)). Then, one fits $\tilde{Q}^\pi_w$ based on $Q^\pi_w$. In particular, given transitions $(s, a, r, s')$ sampled from interactions with the environment, one can train $Q^\pi_w$ to minimize the Bellman error $(Q^\pi_w(s, a) - r - \gamma Q^\pi_w(s', a'))^2$ where $a' \sim N(\mu(s'), \Sigma(s'))$. Then, $\tilde{Q}^\pi_w$ can be optimized to minimize the squared error $(\tilde{Q}^\pi_w(s, a) - \mathbb{E}_{\tilde{a}} Q^\pi_w(s, \tilde{a}))^2$ where $\tilde{a} \sim N(a, \Sigma(s))$, using several samples. When the target values in these residuals are treated as fixed (*i.e.,* using a target network), such a training procedure will achieve a fixed point when $\tilde{Q}^\pi_w(s, a)$ satisfies the recursion in the Bellman equation (10).

The second approach requires a single function approximator for $\tilde{Q}^\pi_w(s, a)$, resulting in a simpler implementation, and thus we use this approach in our experimental evaluation. Suppose one has access to a tuple $(s, \tilde{a}, \tilde{r}, \tilde{s}')$ sampled from a replay buffer with knowledge of the sampling probability $q(\tilde{a} \mid s)$ (possibly unnormalized). Then assuming that this sampling distribution has a full support, we draw a *phantom* action $a \sim N(\tilde{a}, \Sigma(s))$ and optimize $\tilde{Q}^\pi_w(s, a)$ by minimizing a weighted Bellman error $\frac{1}{q(\tilde{a}|s)}(\tilde{Q}^\pi_w(s, a) - \tilde{r} - \gamma \tilde{Q}^\pi_w(\tilde{s}', \mu(\tilde{s}')))^2$. For a specific pair of state and action $(s, a)$ the

expected value of the objective is,

$$E(w \mid (s, a)) = \mathbb{E}_{q(\tilde{a} \mid s), \tilde{r}, \tilde{s}'} \left[ \frac{N(a \mid \tilde{a}, \Sigma(s))}{q(\tilde{a} \mid s)} (\tilde{Q}_w^\pi(s, a) - \tilde{r} - \gamma \tilde{Q}_w^\pi(\tilde{s}', \mu(\tilde{s}')))^2 \right]. \qquad (17)$$

Note that $N(a|\tilde{a}, \Sigma(s)) = N(\tilde{a}|a, \Sigma(s))$. Therefore, when the target value $\tilde{r} + \gamma \tilde{Q}_w^\pi(\tilde{s}', \mu(\tilde{s}'))$ is treated as fixed (*e.g.,* when using target networks) this training procedure reaches an optimum when $\tilde{Q}_w^\pi(s, a)$ satisfies the recursion in the Bellman equation (13).

In practice, we find that it is unnecessary to keep track of the probabilities $q(\tilde{a} \mid s)$, and assume the replay buffer provides a near-uniform distribution of actions conditioned on states. Other recent work has also benefited from ignoring or heavily damping importance weights (Munos et al., 2016; Wang et al., 2017; Schulman et al., 2017). However, it is possible when interacting with the environment to save the probability of sampled actions along with their transitions, and thus have access to $q(\tilde{a} \mid s) \approx N(\tilde{a} \mid \mu_{\text{old}}(s), \Sigma_{\text{old}}(s))$.

### 3.3 PROXIMAL POLICY OPTIMIZATION

Policy gradient algorithms are notoriously unstable, particularly in continuous control problems. Such instability has motivated the development of trust region methods that attempt to mitigate the issue by constraining each gradient step to lie within a trust region (Schulman et al., 2015), or augmenting the expected reward objective with a penalty on KL-divergence from a previous policy (Nachum et al., 2017b; Schulman et al., 2017; Azar et al., 2012). These stabilizing techniques have thus far not been applicable to algorithms like DDPG, since the policy is deterministic. The formulation we propose in this paper, however, is easily amenable to trust region optimization. Specifically, we may augment the objective (11) with a penalty

$$O_{\text{TR}}(\pi) = O_{\text{ER}}(\pi) - \lambda \int_{\mathcal{S}} \text{KL}\left(\pi \parallel \pi_{\text{old}}\right) d\rho^\pi(s), \qquad (18)$$

where $\pi_{\text{old}} \equiv (\mu_{\text{old}}, \Sigma_{\text{old}})$ is a previous parameterization of the policy. The optimization is straightforward, since the KL-divergence of two Gaussians can be expressed analytically.

## 4 RELATED WORK

This paper follows a long line of work that uses Q-value functions to stably learn a policy, which in the past has been used to either approximate expected (Rummery & Niranjan, 1994; Van Seijen et al., 2009; Gu et al., 2017) or optimal (Watkins, 1989; Silver et al., 2014; Nachum et al., 2017a; Haarnoja et al., 2017; Metz et al., 2017) future value.

Work that is most similar to what we present are methods that exploit gradient information from the Q-value function to train a policy. Deterministic policy gradient (Silver et al., 2014) is perhaps the best known of these. The method we propose can be interpreted as a generalization of the deterministic policy gradient. Indeed, if one takes the limit of the policy covariance $\Sigma(s)$ as it goes to 0, the proposed Q-value function becomes the deterministic value function of DDPG, and the updates for training the Q-value approximator and the policy mean are identical.

Stochastic Value Gradient (SVG) (Heess et al., 2015) also trains stochastic policies using an update that is similar to DDPG (*i.e.,* SVG(0) with replay). The key differences with our approach are that SVG does not provide an update for the covariance, and the mean update in SVG estimates the gradient with a noisy Monte Carlo sample, which we avoid by estimating the smoothed Q-value function. Although a covariance update could be derived using the same reparameterization trick as in the mean update, that would also require a noisy Monte Carlo estimate. Methods for updating the covariance along the gradient of expected reward are essential for applying the subsequent trust region and proximal policy techniques.

More recently, Ciosek & Whiteson (2017) introduced expected policy gradients (EPG), a generalization of DDPG that provides updates for the mean and covariance of a stochastic Gaussian policy using gradients of an estimated Q-value function. In that work, the expected Q-value used in standard policy gradient algorithms such as SARSA (Sutton & Barto, 1998; Rummery & Niranjan, 1994; Van Seijen et al., 2009) is estimated. The updates in EPG therefore require approximating

an integral of the expected Q-value function. Our analogous process directly estimates an integral (via the smoothed Q-value function) and avoids approximate integrals, thereby making the updates simpler. Moreover, while Ciosek & Whiteson (2017) rely on a quadratic Taylor expansion of the estimated Q-value function, we instead rely on the strength of neural network function approximators to directly estimate the smoothed Q-value function.

The novel training scheme we propose for learning the covariance of a Gaussian policy relies on properties of Gaussian integrals (Bonnet, 1964; Price, 1958). Similar identities have been used in the past to derive updates for variational auto-encoders (Kingma & Welling, 2014) and Gaussian back-propagation (Rezende et al., 2014).

Finally, the perspective presented in this paper, where Q-values represent the averaged return of a distribution of actions rather than a single action, is distinct from recent advances in distributional RL (Bellemare et al., 2017). Those approaches focus on the distribution of returns of a single action, whereas we consider the single average return of a distribution of actions. Although we restrict our attention in this paper to Gaussian policies, an interesting topic for further investigation is to study the applicability of this new perspective to a wider class of policy distributions.

## 5 EXPERIMENTS

We utilize the insights from Section 3 to introduce a new RL algorithm, *Smoothie*. Smoothie maintains a parameterized $\tilde{Q}_w^\pi$ trained via the procedure described in Section 3.2. It then uses the gradient and Hessian of this approximation to train a Gaussian policy $\mu_\theta, \Sigma_\phi$ using the updates stated in (14) and (16). See Algorithm 1 for a simplified pseudocode of our algorithm.

---

**Algorithm 1** Smoothie

---

**Input:** Environment $ENV$, learning rates $\eta_\pi, \eta_Q$, discount factor $\gamma$, KL-penalty $\lambda$, batch size $B$, number of training steps $N$, target network lag $\tau$.

Initialize $\theta, \phi, w$, set $\theta' = \theta, \phi' = \phi, w' = w$.
**for** $i = 0$ **to** $N - 1$ **do**
    *// Collect experience*
    Sample action $a \sim N(\mu_\theta(s), \Sigma_\phi(s))$ and apply to $ENV$ to yield $r$ and $s'$.
    Insert transition $(s, a, r, s')$ to replay buffer.

    *// Train $\mu, \Sigma$*
    Sample batch $\{(s_k, a_k, r_k, s'_k)\}_{k=1}^B$ from replay buffer.
    Compute gradients $g_k = \frac{\partial \tilde{Q}_w^\pi(s_k, a)}{\partial a}\big|_{a=\mu_\theta(s_k)}$.
    Compute Hessians $H_k = \frac{\partial^2 \tilde{Q}_w^\pi(s_k, a)}{\partial a^2}\big|_{a=\mu_\theta(s_k)}$.
    Compute KL-penalties $KL_k = KL(\mu_\theta, \Sigma_\phi || \mu_{\theta'}, \Sigma_{\phi'})$.
    Compute updates
        $\Delta\theta = \frac{1}{B}\sum_{k=1}^B g_k \nabla_\theta \mu_\theta(s_k) - \lambda\nabla_\theta KL_k$,
        $\Delta\phi = \frac{1}{B}\sum_{k=1}^B \frac{1}{2}H_k \nabla_\phi \Sigma_\phi(s_k) - \lambda\nabla_\phi KL_k$.
    Update $\theta \leftarrow \theta + \eta_\pi\Delta\theta, \phi \leftarrow \phi + \eta_\pi\Delta\phi$.

    *// Train $\tilde{Q}^\pi$*
    Sample batch $\{(s_k, \tilde{a}_k, \tilde{r}_k, \tilde{s}'_k)\}_{k=1}^B$ from replay buffer.
    Sample phantom actions $a_k \sim N(\tilde{a}_k, \Sigma_\phi(s_k))$.
    Compute loss $\mathcal{L}(w) = \frac{1}{B}\sum_{k=1}^B (\tilde{Q}_w^\pi(s, a) - r - \gamma\tilde{Q}_{w'}^\pi(\tilde{s}', \mu_{\theta'}(\tilde{s}')))^2$.
    Update $w \leftarrow w - \eta_Q\nabla_w\mathcal{L}(w)$.

    *// Update target variables*
    Update $\theta' \leftarrow (1-\tau)\theta' + \tau\theta, \phi' \leftarrow (1-\tau)\phi' + \tau\phi, w' \leftarrow (1-\tau)w' + \tau w$.
**end for**

---

We perform a number of evaluations of Smoothie compared to DDPG. We choose DDPG as a baseline because it (1) utilizes gradient information of a Q-value approximator, much like our algorithm; and (2) is a standard algorithm well-known to have achieve good, sample-efficient performance on continuous control benchmarks.

## 5.1 A SYNTHETIC TASK

To evaluate Smoothie we begin with a simple synthetic task which allows us to study its behavior in a restricted setting. We devised a simple single-action one-shot environment in which the reward function is a mixture of two Gaussians, one better than the other (see Figure 1 (Right)). We initialize the policy mean to be centered on the worse of the two Gaussians. We plot the learnable policy mean and standard deviation during training for Smoothie and DDPG in Figure 1 (Left). Smoothie learns both the mean and variance, while DDPG learns only the mean and the variance plotted is the exploratory noise, whose scale is kept fixed during training.

As expected we observe that DDPG cannot escape the local optimum. At the beginning of training it exhibits some movement away from the local optimum (likely due to the initial noisy approximation given by $Q_w^\pi$), it is unable to progress very far from the initial mean. Note that this is not an issue of exploration. The exploration scale is high enough that $Q_w^\pi$ is aware of the better Gaussian. The issue is in the update for $\mu_\theta$, which is only with regard to the derivative of $Q_w^\pi$ at the current mean.

On the other hand, we find Smoothie is successfully able to solve the task. This is because the smoothed reward function approximated by $\tilde{Q}_w^\pi$ has a derivative which clearly points $\mu_\theta$ towards the better Gaussian. We also observe that Smoothie is able to suitably adjust the covariance $\Sigma_\phi$ during training. Initially, $\Sigma_\phi$ decreases due to the concavity of the smoothed reward function. As a region of convexity is entered, it begins to increase, before again decreasing to near-zero as $\mu_\theta$ approaches the global optimum.

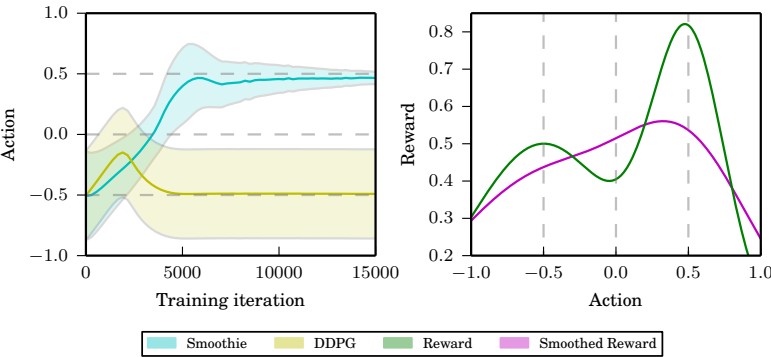

Figure 1: Left: The learnable policy mean and standard deviation during training for Smoothie and DDPG on a simple one-shot synthetic task. The standard deviation for DDPG is the exploratory noise kept constant during training. Right: The reward function for the synthetic task along with its Gaussian-smoothed version. We find that Smoothie can successfully escape the lower-reward local optimum. We also notice Smoothie increases and decreases its policy variance as the convexity/concavity of the smoothed reward function changes.

## 5.2 CONTINUOUS CONTROL

We now turn our attention to standard continuous control benchmarks available on OpenAI Gym (Brockman et al., 2016) utilizing the MuJoCo environment (Todorov et al., 2012).

Our implementations utilize feed forward neural networks for policy and Q-values. We parameterize the covariance $\Sigma_\phi$ as a diagonal given by $e^\phi$. The exploration for DDPG is determined by an Ornstein-Uhlenbeck process (Uhlenbeck & Ornstein, 1930; Lillicrap et al., 2016). Additional implementation details are provided in the Appendix.

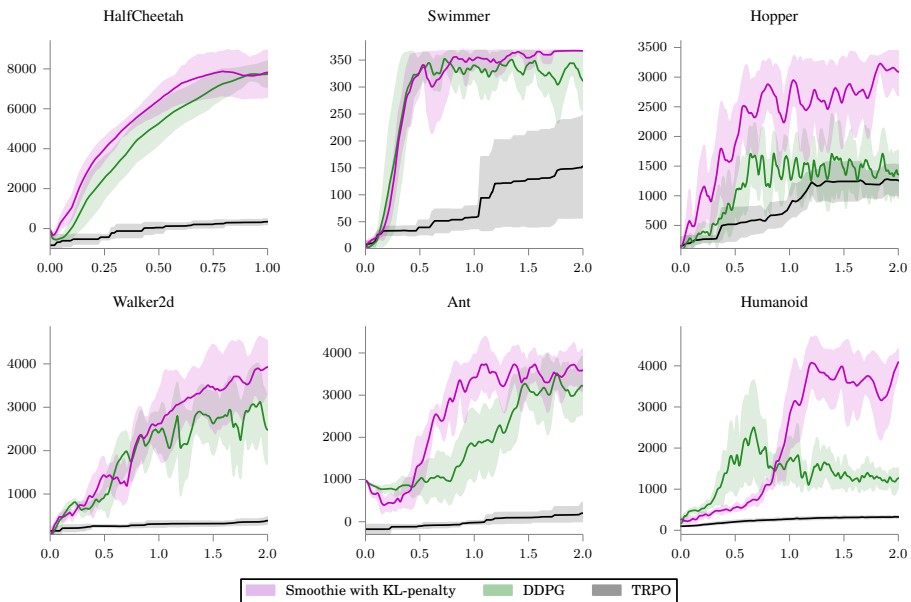

Figure 2: Results of Smoothie, DDPG, and TRPO on continuous control benchmarks. The x-axis is in millions of environment steps. Each plot shows the average reward and standard deviation clipped at the min and max of six randomly seeded runs after choosing best hyperparameters. We see that Smoothie is competitive with DDPG even when DDPG uses a hyperparameter-tuned noise scale, and Smoothie learns the optimal noise scale (the covariance) during training. Moreoever, we observe significant advantages in terms of final reward performance, especially in the more difficult tasks like Hopper, Walker2d, and Humanoid. Across all tasks, TRPO is not sufficiently sample-efficient to provide a competitive baseline.

We compare the results of Smoothie and DDPG in Figure 2. For each task we performed a hyperparameter search over actor learning rate, critic learning rate and reward scale, and plot the average of six runs for the best hyperparameters. For DDPG we extended the hyperparameter search to also consider the scale and damping of exploratory noise provided by the Ornstein-Uhlenbeck process. Smoothie, on the other hand, contains an additional hyperparameter to determine the weight on KL-penalty.

Despite DDPG having the advantage of its exploration decided by a hyperparameter search while Smoothie must learn its exploration without supervision, we find that Smoothie performs competitively or better across all tasks, exhibiting a slight advantage in Swimmer and Ant, while showing more dramatic improvements in Hopper, Walker2d, and Humanoid. The improvement is especially dramatic for Hopper, where the average reward is doubled. We also highlight the results for Humanoid, which as far as we know, are the best published results for a method that only trains on the order of millions of environment steps. In contrast, TRPO, which to the best of our knowledge is the only other algorithm which can achieve better performance, requires on the order of tens of millions of environment steps to achieve comparable reward. This gives added evidence to the benefits of using a learnable covariance and not restricting a policy to be deterministic.

Empirically, we found the introduction of a KL-penalty to improve performance of Smoothie, especially on harder tasks. We present a comparison of results of Smoothie with and without the KL-penalty on the four harder tasks in Figure 3. A KL-penalty to encourage stability is not possible in DDPG. Thus, our algorithm provides a much needed solution to the inherent instability in DDPG training.

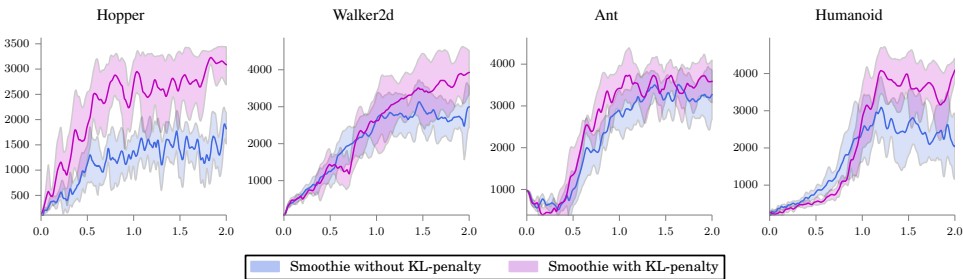

Figure 3: Results of Smoothie with and without a KL-penalty. The x-axis is in millions of environment steps. We observe benefits of using a proximal policy optimization method, especially in Hopper and Humanoid, where the performance improvement is significant without sacrificing sample efficiency.

## 6 CONCLUSION

We have presented a new Q-value function, $\tilde{Q}^\pi$, that is a Gaussian-smoothed version of the standard expected Q-value, $Q^\pi$. The advantage of using $\tilde{Q}^\pi$ over $Q^\pi$ is that its gradient and Hessian possess an intimate relationship with the gradient of expected reward with respect to mean and covariance of a Gaussian policy. The resulting algorithm, Smoothie, is able to successfully learn both mean and covariance during training, leading to performance that can match or surpass that of DDPG, especially when incorporating a penalty on divergence from a previous policy.

The success of $\tilde{Q}^\pi$ is encouraging. Intuitively it may be argued that learning $\tilde{Q}^\pi$ is more sensible than learning $Q^\pi$. The smoothed Q-values by definition make the true reward surface smoother, thus possibly easier to learn; moreover the smoothed Q-values have a more direct relationship with the expected discounted return objective. We encourage future work to further investigate these claims as well as techniques to apply the underlying motivations for $\tilde{Q}^\pi$ to other types of policies.

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

## A    PROOF OF EQUATION (15)

We note that similar identities for Gaussian integrals exist in the literature (Price, 1958; Rezende et al., 2014) and point the reader to these works for further information.

The specific identity we state may be derived using standard matrix calculus. We make use of the fact that

$$\frac{\partial}{\partial A}|A|^{-1/2} = -\frac{1}{2}|A|^{-3/2}\frac{\partial}{\partial A}|A| = -\frac{1}{2}|A|^{-1/2}A^{-1}, \tag{19}$$

and for symmetric $A$,

$$\frac{\partial}{\partial A}||v||^2_{A^{-1}} = -A^{-1}vv^T A^{-1}. \tag{20}$$

We omit $s$ from $\Sigma(s)$ in the following equations for succinctness. The LHS of (15) is

$$\int_{\mathcal{A}} Q^\pi(s,\tilde{a})\frac{\partial}{\partial\Sigma}N(\tilde{a}|a,\Sigma)\mathrm{d}\tilde{a}$$

$$= \int_{\mathcal{A}} Q^\pi(s,\tilde{a})\exp\left\{-\frac{1}{2}||\tilde{a}-a||^2_{\Sigma^{-1}}\right\}\left(\frac{\partial}{\partial\Sigma}|2\pi\Sigma|^{-1/2} - \frac{1}{2}|2\pi\Sigma|^{-1/2}\frac{\partial}{\partial\Sigma}||\tilde{a}-a||^2_{\Sigma^{-1}}\right)\mathrm{d}\tilde{a}$$

$$= \frac{1}{2}\int_{\mathcal{A}} Q^\pi(s,\tilde{a})N(\tilde{a}|a,\Sigma)\left(-\Sigma^{-1} + \Sigma^{-1}(\tilde{a}-a)(\tilde{a}-a)^T\Sigma^{-1}\right)\mathrm{d}\tilde{a}. \tag{21}$$

Meanwhile, towards tackling the RHS of (15) we note that

$$\frac{\partial\tilde{Q}^\pi(s,a)}{\partial a} = \int_{\mathcal{A}} Q^\pi(s,\tilde{a})N(\tilde{a}|a,\Sigma)\Sigma^{-1}(\tilde{a}-a)\mathrm{d}\tilde{a}. \tag{22}$$

Thus we have

$$\frac{\partial^2\tilde{Q}^\pi(s,a)}{\partial a^2} = \int_{\mathcal{A}} Q^\pi(s,\tilde{a})\left(\Sigma^{-1}(\tilde{a}-a)\frac{\partial}{\partial a}N(\tilde{a}|a,\Sigma) + N(\tilde{a}|a,\Sigma)\frac{\partial}{\partial a}\Sigma^{-1}(\tilde{a}-a)\right)\mathrm{d}\tilde{a} \tag{23}$$

$$= \int_{\mathcal{A}} Q^\pi(s,\tilde{a})N(\tilde{a}|a,\Sigma)(\Sigma^{-1}(\tilde{a}-a)(\tilde{a}-a)^T\Sigma^{-1} - \Sigma^{-1})\mathrm{d}\tilde{a}. \tag{24}$$

∎

## B    COMPATIBLE FUNCTION APPROXIMATION

A function approximator $\tilde{Q}^\pi_w$ of $\tilde{Q}^\pi$ should be sufficiently accurate so that updates for $\mu_\theta, \Sigma_\phi$ are not affected by substituting $\frac{\partial\tilde{Q}^\pi_w(s,a)}{\partial a}$ and $\frac{\partial^2\tilde{Q}^\pi_w(s,a)}{\partial a^2}$ for $\frac{\partial\tilde{Q}^\pi(s,a)}{\partial a}$ and $\frac{\partial^2\tilde{Q}^\pi(s,a)}{\partial a^2}$, respectively.

We claim that a $\tilde{Q}^\pi_w$ is compatible with respect to $\mu_\theta$ if

1. $\nabla_a\tilde{Q}^\pi_w(s,a)\big|_{a=\mu_\theta(s)} = \nabla_\theta\mu_\theta(s)^T w$,

2. $\nabla_w\int_{\mathcal{S}}\left(\nabla_a\tilde{Q}^\pi_w(s,a)\big|_{a=\mu_\theta(s)} - \nabla_a\tilde{Q}^\pi(s,a)\big|_{a=\mu_\theta(s)}\right)^2 \mathrm{d}\rho^\pi(s) = 0$ (*i.e.*, $w$ minimizes the expected squared error of the gradients).

Additionally, $\tilde{Q}^\pi_w$ is compatible with respect to $\Sigma_\phi$ if

1. $\nabla^2_a\tilde{Q}^\pi_w(s,a)\big|_{a=\mu_\theta(s)} = \nabla_\phi\Sigma_\phi(s)^T w$,

2. $\nabla_w\int_{\mathcal{S}}\left(\nabla^2_a\tilde{Q}^\pi_w(s,a)\big|_{a=\mu_\theta(s)} - \nabla^2_a\tilde{Q}^\pi(s,a)\big|_{a=\mu_\theta(s)}\right)^2 \mathrm{d}\rho^\pi(s) = 0$ (*i.e.*, $w$ minimizes the expected squared error of the Hessians).

One possible parameterization of $\tilde{Q}^\pi_w$ may be achieved by taking $w = [w_0, w_1, w_2]$ and parameterizing

$$\tilde{Q}^\pi_w(s,a) = V_{w_0}(s) + (a-\mu_\theta(s))^T\nabla_\theta\mu_\theta(s)^T w_1 + (a-\mu_\theta(s))^T\nabla_\phi\Sigma_\phi(s)^T w_2(a-\mu_\theta(s)). \tag{25}$$

| Hyperparameter | Range | Sampling |
|---|---|---|
| actor learning rate | [1e-6,1e-3] | log |
| critic learning rate | [1e-6,1e-3] | log |
| reward scale | [0.01,0.3] | log |
| OU damping | [1e-4,1e-3] | log |
| OU stddev | [1e-3,1.0] | log |
| $\lambda$ | [1e-6, 4e-2] | log |
| discount factor | 0.995 | fixed |
| target network lag | 0.01 | fixed |
| batch size | 128 | fixed |
| clipping on gradients of $Q$ | 4.0 | fixed |
| num gradient updates per observation | 1 | fixed |
| Huber loss clipping | 1.0 | fixed |

Table 1: Random hyperparameter search procedure. We also include the hyperparameters which we kept fixed.

**Proof.** We shall show how the conditions stated for compatibility with respect to $\Sigma_\phi$ are sufficient. The reasoning for $\mu_\theta$ follows via a similar argument. We also refer the reader to Silver et al. (2014) which includes a similar procedure for showing compatibility.

From the second condition for compatibility with respect to $\Sigma_\phi$ we have

$$\int_{\mathcal{S}} \left( \nabla_a^2 \tilde{Q}_w^\pi(s,a)\big|_{a=\mu_\theta(s)} - \nabla_a^2 \tilde{Q}^\pi(s,a)\big|_{a=\mu_\theta(s)} \right) \nabla_w \left( \nabla_a^2 \tilde{Q}_w^\pi(s,a)\big|_{a=\mu_\theta(s)} \right) \mathrm{d}\rho^\pi(s) = 0. \quad (26)$$

We may combine this with the first condition to find

$$\int_{\mathcal{S}} \nabla_a^2 \tilde{Q}_w^\pi(s,a)\big|_{a=\mu_\theta(s)} \nabla_\phi \Sigma_\phi(s) \mathrm{d}\rho^\pi(s) = \int_{\mathcal{S}} \nabla_a^2 \tilde{Q}^\pi(s,a)\big|_{a=\mu_\theta(s)} \nabla_\phi \Sigma_\phi(s) \mathrm{d}\rho^\pi(s), \quad (27)$$

which is the desired property for compatibility. ∎

While it is reassuring to know that there exists a class of function approximators which are compatible, this fact is largely ignored in practice. Not only is the class of compatible functions heavily restricted in terms of expressiveness, due to the first set of conditions for $\mu_\theta, \Sigma_\phi$, it is also impossible to satisfy the second set of conditions without access to derivative and Hessian information of the true $\tilde{Q}^\pi$. This problem is also present in DDPG, and we feel this issue merits additional investigation in future work.

## C  IMPLEMENTATION DETAILS

We utilize feed forward networks for both policy and Q-value approximator. For $\mu_\theta(s)$ we use two hidden layers of dimensions $(400, 300)$ and relu activation functions. For $\tilde{Q}_w^\pi(s,a)$ and $Q_w^\pi(s,a)$ we first embed the state into a 400 dimensional vector using a fully-connected layer and $\tanh$ non-linearity. We then concatenate the embedded state with $a$ and pass the result through a 1-hidden layer neural network of dimension 300 with $\tanh$ activations. We use a diagonal $\Sigma_\phi(s) = e^\phi$ for Smoothie, with $\phi$ initialized to $-1$.

To find optimal hyperparameters we perform a 100-trial random search over the hyperparameters specified in Table 1. The OU exploration parameters only apply to DDPG. The $\lambda$ coefficient on KL-penalty only applies to Smoothie with a KL-penalty.

