# OpenReview forum: "Learning Gaussian Policies from Smoothed Action Value Functions"
_ICLR.cc/2018/Conference — Reject_

### Official Review · AnonReviewer1 · 2017-11-27
**seems to be a good paper, however, I do not even see an exact algorithm formulation**

**Rating:** 6
**Confidence:** 4

**Review:**

I think I should understand the gist of the paper, which is very interesting, where the action of \tilde Q(s,a) is drawn from a distribution. The author also explains in detail the relation with PGQ/Soft Q learning, and the recent paper "expected policy gradient" by Ciosek & Whiteson. All these seems very sound and interesting.

Weakness:
1. The major weakness is that throughout the paper, I do not see an algorithm formulation of the Smoothie algorithm, which is the major algorithmic contribution of the paper (I think the major contribution of the paper is on the algorithmic side instead of theoretical). Such representation style is highly discouraging and brings about un-necessary readability difficulties.

2. Sec. 3.3 and 3.4 is a little bit abbreviated from the major focus of the paper, and I guess they are not very important and novel (just educational guess, because I can only guess what the whole algorithm Smoothie is). So I suggest moving them to the Appendix and make the major focus more narrowed down.

---

> ### Author Response · Authors · 2018-01-05
> **Response**
>
> We thank the reviewer for their valuable feedback.
>
> R1: "The major weakness is that throughout the paper, I do not see an algorithm formulation of the Smoothie algorithm, which is the major algorithmic contribution of the paper … Such representation style is highly discouraging and brings about un-necessary readability difficulties."
>
> We take the presentation criticism seriously. To improve the exposition, we have updated the paper to include an algorithm box with a pseudo-code description of the implementation.
>
> R1: "I think the major contribution of the paper is on the algorithmic side instead of theoretical"
>
> Not entirely.  Note that the derived updates for the mean and covariance parameters of a Gaussian policy in terms of the gradient and Hessian of smoothed Q-values are novel. Also, the relation between the Hessian and the covariance update (Eq. 15) is particularly novel; we are not aware of any similar equations previously used in RL.
>
> R1: "Sec. 3.3 and 3.4 is a little bit abbreviated from the major focus of the paper, and I guess they are not very important and novel (just educational guess, because I can only guess what the whole algorithm Smoothie is). So I suggest moving them to the Appendix and make the major focus more narrowed down.”
>
> Thank you for the suggestion. We agree that Section 3.3 is a theoretical aside, which may not interest most readers. We have updated the paper to move Section 3.3 to the appendix, leaving more room in the main body for the pseudo-code presentation. On the other hand, we believe Section 3.4 is important as it explains the specific technique which yields the best empirical performance.

---

### Official Review · AnonReviewer2 · 2017-11-27
**The paper introduces smoothed Q-values, a tweak on standard Q-functions. It demonstrates some nice theoretical properties and reasonably successful experiments. The paper is interesting and correct, but unlikely to make a big impact.**

**Rating:** 6
**Confidence:** 4

**Review:**

The paper introduces smoothed Q-values, defined as the value of drawing an action from a Gaussian distribution and following a given policy thereafter.  It demonstrates that this formulation can still be optimized with policy gradients, and in fact is able to dampen instability in this optimization using the KL-divergence from a previous policy, unlike preceding techniques.  Experiments are performed on an simple domain which nicely demonstrates its properties, as well as on continuous control problems, where the technique outperforms or is competitive with DDPG.

The paper is very clearly written and easy to read, and its contributions are easy to extract.  The appendix is quite necessary for the understanding of this paper, as all proofs do not fit in the main paper.  The inclusion of proof summaries in the main text would strengthen this aspect of the paper.

On the negative side, the paper fails to make a strong case for significant impact of this work; the solution to this, of course, is not overselling benefits, but instead having more to say about the approach or finding how to produce much better experimental results than the comparative techniques.  In other words, the slightly more stable optimization and slightly smaller hyperparameter search for this approach is unlikely to result in a large impact.

Overall, however, I found the paper interesting, readable, and the technique worth thinking about, so I recommend its acceptance.

---

> ### Author Response · Authors · 2018-01-05
> **Response**
>
> We thank the reviewer for their valuable feedback.
>
> R2: “The appendix is quite necessary for the understanding of this paper, as all proofs do not fit in the main paper.  The inclusion of proof summaries in the main text would strengthen this aspect of the paper.”
>
> Thank you for the suggestion. We have updated the text to include proof summaries.
>
> R2: “On the negative side, the paper fails to make a strong case for significant impact of this work; the solution to this, of course, is not overselling benefits, but instead having more to say about the approach or finding how to produce much better experimental results than the comparative techniques. In other words, the slightly more stable optimization and slightly smaller hyperparameter search for this approach is unlikely to result in a large impact.”
>
> To demonstrate the significance of our experimental results more clearly, we have updated Figure 2 to compare the performance of Smoothie with KL-penalty, DDPG, and TRPO on continuous control benchmarks. Figure 2 makes it clear that Smoothie achieves the state-of-the-art by converging faster and/or achieving better final rewards. On the challenging Hopper and Humanoid tasks, Smoothie achieves double the average reward compared to DDPG without sacrificing sample efficiency. Our previous presentation of the results in two separate figures showing the difference between Smoothie without KL penalty and DDPG, and between Smoothie with and without the KL penalty made the significance of our results less clear.

---

### Official Review · AnonReviewer3 · 2017-12-05
**This doesn't seem to actually work**

**Rating:** 5
**Confidence:** 3

**Review:**

This paper explores the idea of using policy gradients to learn a stochastic policy on complex control problems.  The central idea is to frame learning in terms of a new kind of Q-value that attempts to smooth out Q-values by framing them in terms of expectations over Gaussian policies.

To be honest, I didn't really "get" this paper.
* As far I understand, all of the original work policy gradients involved stochastic policies.  Many are/were Gaussian.
* All Q-value estimators are designed to marginalize out the randomness in these stochastic policies.
* As far as I can tell, this is equivalent to a slightly different formulation, where the agent emits a deterministic action (\mu,\Sigma) and the environment samples an action from that distribution.  In other words, it seems that if we just draw the box a bit differently, the environment soaks up the nondeterminism, instead of needing to define a new type of Q-value.

Ultimately, I couldn't discern /why/ this was a significant advance for RL, or even a meaningful new perspective on classic ideas.

I thought the little 2-mode MOG was a nice example of the premise of the model.

While I may or may not have understood the core technical contribution, I think the experiments can be critiqued: they didn't really seem to work out.  Figures 2&3 are unconvincing - the differences do not appear to be statistically significant.  Also, I was disappointed to see that the authors only compared to DDPG; they could have at least compared to TRPO, which they mention.  They dismiss it by saying that it takes 10 times as long, but gets a better answer - to which I respond, "Very well, run your algorithm 10x longer and see where you end up!"  I think we need to see a more compelling demonstration of why this is a useful idea before it's ready to be published.

The idea of penalizing a policy based on KL-divergence from a reference policy was explored at length by Bert Kappen's work on KL-MDPs.  Perhaps you should cite that?

---

> ### Author Response · Authors · 2018-01-05
> **Response**
>
> We thank the reviewer for their valuable feedback.
>
> R3: “To be honest, I didn't really "get" this paper.”
>
> We hope that our changes to the paper and the rebuttal make the contributions of the paper clearer.
>
> R3: “As far I understand, all of the original work policy gradients involved stochastic policies.  Many are/were Gaussian.”
>
> Indeed, most of the original work on policy gradients uses stochastic policies. In such a setting, Q-value or value functions are used for variance reduction when estimating the policy gradient; that is, the policy is trained using a form similar to Eq. (4) in the paper.
>
> However, more recently, several algorithms have been proposed (e.g., DDPG, SVG), which use Q-values in a different way. They use the gradient of a Q-function approximator to train the policy. This results in a large improvement in sample efficiency over traditional policy gradient methods.  The most widely used of these algorithms, DDPG, is restricted to deterministic policies. Our work extends DDPG to general Gaussian policies, showing that 1) we can directly learn the smoothed Q-values to avoid estimating an additional Monte Carlo sampling step necessary for SVG, 2) the gradient and the Hessian of the smoothed Q-values can be used to update the mean and the covariance parameters of a Gaussian policy. Notably, although SVG uses a stochastic policy, it uses a fixed covariance.
>
> R3: “All Q-value estimators are designed to marginalize out the randomness in these stochastic policies.”
>
> The smoothed Q-values that we introduce additionally marginalize out the randomness in the first action (a) of a typical Q(s, a) value based on the mean and covariance of the first action. As a result, we avoid an additional Monte Carlo sampling step to draw the first action, as compared to SVG for example.
>
> R3: “As far as I can tell, this is equivalent to a slightly different formulation, where the agent emits a deterministic action (\mu,\Sigma) and the environment samples an action from that distribution. In other words, it seems that if we just draw the box a bit differently, the environment soaks up the nondeterminism, instead of needing to define a new type of Q-value.”
>
> Although one could pursue such an approach, it is not equivalent to the direction we pursue in the paper.  Under the above suggestion, where the agent emits an action (\mu, \Sigma), the corresponding Q-value function would be a function of both \mu and \Sigma. On the other hand, the smoothed Q-value function we consider only takes in \mu. A key contribution of the paper is showing that even though \tilde{Q} is not a direct function of \Sigma, one can still derive an update for \Sigma based on the Hessian of \tilde{Q} with respect to mean action.
>
> R3: “I thought the little 2-mode MOG was a nice example of the premise of the model.”
>
> Thank you.  We hope our responses contribute to a better understanding of the premise of the approach. We expect that this fundamental smoothing behavior is the source of the improvement over DDPG.
>
> R3: “While I may or may not have understood the core technical contribution, I think the experiments can be critiqued: they didn't really seem to work out. Figures 2&3 are unconvincing - the differences do not appear to be statistically significant.”
>
> To demonstrate the significance of our experimental results more clearly, we have updated Figure 2 to compare the performance of Smoothie with KL-penalty, DDPG, and TRPO on continuous control benchmarks. Figure 2 makes it clear that our results are statistically significant and Smoothie achieves the state-of-the-art by converging faster and/or achieving better final rewards. On the challenging Hopper and Humanoid tasks, Smoothie achieves double the average reward compared to DDPG without sacrificing sample efficiency. Our previous presentation of the results in two separate figures showing the difference between Smoothie without KL penalty and DDPG, and between Smoothie with and without the KL penalty made the significance of our results less clear.
>
> R3: “I was disappointed to see that the authors only compared to DDPG; they could have at least compared to TRPO, which they mention.  They dismiss it by saying that it takes 10 times as long, but gets a better answer - to which I respond, "Very well, run your algorithm 10x longer and see where you end up!"
>
> Sample-efficient reinforcement learning (RL) is a key challenge for real world applications of RL, and in this paper we focus on the behavior of the algorithms in the practical data regime. That said, we have included a comparison with TRPO in Figure 2, which shows that in this data regime, TRPO is not competitive. Similar conclusions have been made about TRPO by other papers (e.g. https://arxiv.org/abs/1707.06347).
>
> R3: “The idea of penalizing a policy based on KL-divergence from a reference policy was explored at length by Bert Kappen's work on KL-MDPs.  Perhaps you should cite that?”
>
> Reference added.

---

### Decision · Program_Chairs · 2018-01-29
**ICLR 2018 Conference Acceptance Decision**

**Decision:**

Reject

**Comment:**

Thank you for submitting you paper to ICLR. Two of the reviewers are concerned that the paper’s contributions are not significant enough —either in terms of the theoretical or experimental contribution -- to warrant publication. The authors have improved the experimental aspect to include a more comprehensive comparison, but this has not moved the reviewers.

Summary: The approach is very promising, but more experimental work is still required to demonstrate significance.